# Preparation of Boron/Sulfur-Codoped Porous Carbon Derived from Biological Wastes and Its Application in a Supercapacitor

**DOI:** 10.3390/nano12071182

**Published:** 2022-04-01

**Authors:** Yanbin Wang, Dian Wang, Zhaoxia Li, Qiong Su, Shuai Wei, Shaofeng Pang, Xiangfei Zhao, Lichun Liang, Lihui Kang, Shijun Cao

**Affiliations:** 1School of Chemical Engineering, Northwest Minzu University, Lanzhou 730030, China; ybwang@126.com (Y.W.); d_wang1997@163.com (D.W.); w17803870317@163.com (S.W.); pangshaofeng2006@163.com (S.P.); woaini6704259@163.com (X.Z.); fgl770880llcfgl@163.com (L.L.); lihuikang0325@163.com (L.K.); a2848695227@163.com (S.C.); 2Key Laboratory of Environmentally Friendly Composite Materials of the State Ethnic Affairs Commission, Key Laboratory of Utility of Environmental Friendly Composite Materials and Biomass in Universities of Gansu Province, Lanzhou 730030, China; 3Engineering Research Center of Biomass-Functional Composite Materials of Gansu Province, Lanzhou 730030, China

**Keywords:** biomass, boron/sulfur-codoped, porous carbon, synergetic effect, supercapacitor

## Abstract

Abundant biomass resources are a good choice for preparing electrode materials for supercapacitors, but developing a versatile and simple synthetic method to convert them into electrode materials remains a challenge. In the present research, our team reports a promising strategy and cost-efficient method to fabricate boron/sulfur-codoped porous carbon from biomass sources, mainly utilizing four biomass materials. Detailed material characterization showed that the samples produced by this approach possess rich B and S doping. Additionally, the original biomass materials treated by activation produce abundant pores. Therefore, owing to the synergetic effect of abundant atomic doping and microporous/mesoporous distribution, the obtained carbon as electrode material manifested excellent specific capacitances of 290 F g^−1^ at a 0.5 A g^−1^ current density. Moreover, the specific energy of the prepared samples of the as-assembled symmetric supercapacitor is as high as 16.65 Wh kg^−1^ in 1 M Na_2_SO_4_, with a brilliant cyclical performance of only a 2.91% capacitance decay over 10,000 cycles. In addition, it has been verified universally that three other types of bio-wastes can also prepare electrode material using this method. This paper represents a significant attempt to turn waste biomass into treasure while also providing ideas for the design and preparation of supercapacitor electrode materials.

## 1. Introduction

To face the challenge of the global resource shortage and prevent the over-consumption of fossil fuels, developing sustainable development resources based on clean energy-storing and energy-transformation equipment is regarded as an effective strategy [1,2,3]. Supercapacitors have been viewed as a very promising energy source with the benefits of rapid charging and discharging, great specific energy, and outstanding cyclic performance [4]. In the meantime, their excellent energy-storing capability, long-term stability, and lower maintenance demands make them suitable for a wide range of fields [5]. Currently, various carbon materials, such as graphene, activated charcoal, and carbon nanotubes, have been deeply researched for supercapacitor electrodes [6,7]. However, the complex properties of these carbon materials and the expensive manufacturing costs present a long road to industrialization. Therefore, how to realize low-cost, simpler manufacturing and processing is still a great challenge in practical applications. Carbon materials derived from biomass have been thought of as the currently most promising materials for supercapacitors from both economic and sustainable development points of view [8]. As reported in the literature, many biomass materials have been employed to prepare supercapacitor electrodes, such as soybean [9], kapok flowers shell [10], peach gum [11], pomelo peel [12], and locust seed meal [13].

Unfortunately, most biomass raw materials are often intractable and impure, so they cannot be directly used as electrode materials [14]. Thus, it is crucial to find an appropriate, economical, and generic approach for biochar treatment. Based on their original biological structure, lots of chemical activators are used to create pores, such as KOH [15,16], ZnCl_2_ [17], and H_3_PO_4_ [18]. Generally, the activation mechanism is considered to be an activation process to produce a large number of pore structures and enlarge the active region, which creates a higher diffusion rate and shorter diffusion distance of charge in the ion-transport process [19,20].

On the other hand, the surface chemical properties of the electrode are one of the factors affecting the performance of carbon-based supercapacitors. The transport of electrolyte ions depends on the wettability of carbon materials, but the biochar obtained from wastes is hydrophobic. The atomic doping (such as N, O, B, and S) [21,22,23] is capable of enhancing the surface wettability and providing more electrochemical active reaction spots to improve the interaction between porous carbon frameworks, active sites, and the electrical conductivity of carbon materials [24,25,26]. Moreover, heteroatoms can offer additional pseudocapacitance contributions to carbon materials [27].

At present, more than two kinds of heteroatomic co-doping have been widely used to improve the properties of carbon materials. Among them, B/S co-doping has been applied in many fields. Wen et al. synthesized B/S co-doped carbon material for the electrochemical nitrogen-reduction reaction (NRR) using carbon nanotubes as a matrix and nanometer reactor. Due to the interaction of B/S elements, the heteroatoms doping of the S atom resulted in a change in the position of the pz’s orbital center of the B atom so that the heteroatoms catalyst could effectively strengthen the NRR activity [28]. Tian et al. prepared S/B-rGO composites as cathode material for Li/S batteries by a simple method. They found that the introduction of B and S elements enhanced the conductivity of the composite cathode and delayed the shuttle effect, displaying good cycling stability and rate capacity [29]. Nevertheless, there are few reports about B/S co-doped carbon materials employed in supercapacitors, except for combinations such as B/N and N/S, which are widely developed for supercapacitors. According to relevant studies, B is a unique element that can act as an electrocatalyst to form an active carbon surface and promote the redox reaction of N and O functional groups [23,30,31]. In addition, the polarity of the B atom is positive, so it can chemically adsorb negative charge on the carbon skeleton surface to provide additional capacitance. S atom doping destroys the equilibrium density of electrons, which efficiently ameliorates the hydrophilicity of the carbon surface [32,33,34].

H_3_BO_3_ is a green boron dopant because of its low toxicity. It is envisaged that B atoms are introduced into biomass carbon using boric acid as a dopant. Additionally, it has been reported that when S and B are co-doped, the chemical interaction will be enhanced; additionally, the content of B will be raised [29], thereby enhancing electrochemical performance. In this paper, boron/sulfur co-doped porous carbon (B/S-SC_S_-1) was prepared by using the ZnCl_2_ activation process using boric acid and thiourea as B and S sources and sedum stem as raw materials for the electrode material of the supercapacitor. The prepared samples showed porous structures with interconnections and an abundant surface heteroatomic content. When only boron was doped, the content of the B atom was only 0.61%. While B/S was co-doped, the B content increased to 5.56%. When utilized as the electrode material for a supercapacitor in 6 M KOH, B/S-SC_S_-1 realizes a specific capacitance of 290.7 F g^−1^ at 0.5 A g^−1^, and the 1 M Na_2_SO_4_ utilized as the electrolyte in the assembled B/S-SC_S_-1//B/S-SC_S_-1 symmetric supercapacitor showed great specific energy of 16.65 Wh kg^−1^ at 450 W kg^−1^. To explore the universality of this program, the same scheme was used to synthesize porous carbon with three other types of biomass wastes—walnut peel, wheat straw, and corn stalk—to verify the versatility of the method. This work provides a reference for the more convenient and lower-cost synthesis of porous carbon from biomass.

## 2. Materials and Methods

### Preparation of Electrode Materials

The schematic for the fabrication of B/S-SC_S_-1 is shown in Figure 1. First, the sedum spectabile stalk was washed with deionized water and then ground into powder. Next, 2 g of H_3_BO_3_ and 2 g of CH_4_N_2_S underwent thrombolysis in 40 mL distilled water. Subsequently, 2 g of ZnCl_2_ and 2 g of sedum spectabile stalk were mixed in the above solution and then dried at 80 °C overnight. Then, the above mixture was transferred to a porcelain boat and annealed at 700 °C for 2 h under nitrogen protection in a tube furnace. Finally, the products were washed with 2 M HCl solution and deionized water and dried at 80 °C overnight to obtain B/S-SC_S_-1. The same schemes and experimental conditions were also applied to walnut peel, wheat straw, and corn stalks.

For comparison studies, raw sedum spectabile stalk carbonized under the same conditions without an activating agent and dopant was adopted as the control sample, namely, SC. The samples carbonized under the same conditions using zinc chloride as the activator were denoted as SCs, and the samples doped with boric acid as single boron were denoted as B-SCs. Using a similar method, thiourea was selected to prepare monosulfur-doped carbon materials, denoted as S-SCs, and the samples with boron doping amounts of 0.5 g and 1.5 g were named B/S-SCs-0.5 and B/S-SCs-1.5.

## 3. Results and Discussion

### 3.1. Morphology and Structural Characterization

#### 3.1.1. Morphological Analysis

As seen in Figure 2a, the SEM images of SC highlight a lumpy structure, with a smooth surface of the sedum spectabile stalk after pyrolysis. Figure 2b,c displays the SEM images of B/S-SC_S_-1 at different magnifications; B/S-SC_S_-1 shows a clearly hierarchical porous structure, facilitating the faster adsorption and diffusion of ions from the electrolyte solution. EDS element mapping images visually exhibited that the carbon, nitrogen, oxygen, boron, and sulfur heteroatoms were uniformly distributed throughout the B/S-SC_S_-1 framework structures in Figure 2d. Figure 2e,f further confirms this highly interconnected network structure. Additionally, the B/S-SC_S_-1 high-resolution TEM images are shown in Figure 2g; lots of white spots indicate the existence of abundant micropores, and partial graphitic structures were identified from the observed lattice fringes (aligned graphite tiers), which were associated with the graphite (002) plane. The outcomes unveiled that those pore-rich carbons consisted of abundant amorphous carbons and a few graphite structures. The micropores and mesopores may be related to ZnCl_2_ activation. The activation of ZnCl_2_ can be explained as the vaporization of ZnCl_2_ at high temperatures. Zinc chloride molecules entered the carbon to act as a skeleton; the carbon polymer was carbonized and deposited on the skeleton. After acid and water washing, ZnCl_2_ was removed to form a well-developed pore structure [35]. This hierarchical porous structure offered an electroconductive net with interconnection and short ionic diffusive paths for fast charge transport and ionic migration, causing better electrochemistry properties. The morphological status of the rest of the three pore-rich carbon materials prepared via the ZnCl_2_ activation and B/S doping approach is presented in Appendix A.

#### 3.1.2. X-ray Diffraction (XRD), Raman, and Nitrogen Adsorption and Desorption Analysis

The X-ray diffractometer (XRD) images of the obtained B/S-SC_S_-1 specimens are presented in Figure 3a. B/S-SC_S_-1 exhibited two representative peaks at about 26° and 43.3°, associated with the diffractions of (002) and (001), respectively, and it showed the amorphous carbon characteristics of the sample and had a large number of irregular graphite crystallites [36]. The (101) peak was relatively weak, indicating that the sample graphitization was not uniform, which may be due to the lower activation temperature used in this system, while the degree of graphitization is related to the activation temperature. Appendix A revealed XRD images of these three pore-rich carbon materials; these three porous carbon materials also had similar structures. The Raman spectrum more clearly showed the difference in the degree of graphitization of the sample because it is more sensitive to the microstructure of the carbon material [37]. As presented in Figure 3b, the D peak (disordered or defective band) and G peak (crystalline graphitic band) of the carbon materials appear at the Raman shifts of 1366 cm^−1^ and 1593 cm^−1^, respectively. The samples all display a higher D peak content than the G peak, implying the existence of a large quantity of sp^3^-bonded carbon atoms [38]. The intensity ratio (I_D_/I_G_) of B/S-SC_S_-1 was calculated to be 1.02, which also demonstrated that the doping of B and S elements formed more flaws in the framework of the carbon material but was beneficial for the ion-diffusion process. Thus, it could help to improve the electrochemical performance of B/S-SC_S_-1.

A high porous density is desired for a perfect electrode, facilitating faster ion transport through the carbon layer. Therefore, the pore size distribution of the surface was studied by a nitrogen adsorption–desorption isotherm. As seen in Figure 3c, B/S-SC_S_-1 shows type-IV isothermal lines. The steep adsorption of N_2_ in the low-pressure range (p/p_0_ < 0.1) indicated the presence of micropores [39]. Further, the curve of B/S-SC_S_-1 increased slowly under great relative pressure (p/p_0_ > 0.4), which showed that B/S-SC_S_-1 also has characteristics of mesopores/macropores. It is worth noting that this is beneficial for buffering the electrolyte in practical applications. Combined with Figure 3d, we more accurately confirmed the presence of an abundant small proportion of narrow mesopores (2~5 nm) [40,41]. The specific surface area of B/S-SC_S_-1 was 360 m^2^ g^−1^, based on BET results.

#### 3.1.3. X-ray Photoelectron Spectrometer (XPS) Analysis

The complete XPS survey scan spectrum of all samples is shown in Figure 4. It can be seen from Figure 4a that SC only contained a lower content of Na and O. When only boron was doped (B-SCs), the content of the B atom was only 0.61%. All B/S-SC_S_ samples demonstrated five peaks corresponding to B 1s, C 1s, N 1s, S 2p, and O 1s, respectively (Figure 4a). As expected, the peaks of B and S rose in the survey XPS spectrum of B/S-SC_S_-1 (Figure 4b), meaning that the boron and sulfur elements were smoothly introduced into the carbon due to the use of boric acid and thiourea. At the same time, the XPS spectrum also had N and O elements that can provide N- and O-containing groups to increase the wettability of the material [42]. The high-quality XPS images of C 1s, N 1s, B 1s, and S 2p of B/S-SC_S_-1 were further analyzed to determine their chemical state (Table 1). The C 1s spectra were fitted by four peaks, centered at 284.6, 285.2, 286.4, and 288.3 ± 0.1 eV, designated as C-C, C-N-B, C-O/C-N, and C=O, respectively (Figure 4c). We can see that in addition to carbon–carbon bonds, it also shows a combination with other elements, in which oxygen functional groups were capable of improving the surface hydrophilicity and increasing the pseudocapacitance. N 1s XPS peak (Figure 4d) can be fitted to four components, indicating that there were four species of nitrogen atoms in the sample, including pyridine-N (N-6, 398.8 eV), pyrrole-N (N-5, 400.1 eV), quaternary-N (N-Q, 401.5 eV), and N-X (402.8 eV). The B 1s spectrum is fitted by three peaks, at 190.2, 191.1, and 192 ± 0.1 eV, which could be designated as B-C_2_-O, BN, and B-C-O_2_, respectively (Figure 4e) [43]. Studies have suggested that the relevant functional groups of B can act as an electrochemical catalyst to promote the redox reaction of N and O so that N and O content in the sample can also provide additional pseudocapacitance [44]. Moreover, the B content of B/S-SCs is 5.56 at.%, which can be attributed to the co-doping effect of B/S. The S 2p spectra in B/S-SC_S_-1 materials are fitted by three peaks based on 164, 165.2, and 168.9 ± 0.1 eV as the center and can be designated as C-S, C-S-C, and C-SO_X_-C, respectively, as shown in Figure 4f. These oxidized S functional groups were expected to enhance the wettability of the electrolyte. To explore the effect of doped heteroatoms on wettability, the contact angle measurement was implemented. As seen in Figure 5, for SC, the contact angle of ~109.2° was greater than 90°, indicating that the wetting of the SC surface was unfavorable. For B/S-SC_S_-1, the contact angle was ~78.1°, indicating a hydrophilic feature, which can favor the full wetting of the active species.

### 3.2. Electrochemical Performance of B/S-SC_S_-1 Materials

The electrochemical properties of all samples were first examined in a 6M KOH electrolyte (Appendix A). The CV curves of all sampled materials exhibited a typical rectangular shape (Appendix A). Obviously, the B/S-SC_S_-1 material had a larger CV area than any other material. Additionally, the GCD curves of all sampled materials at a current density of 1 A g^−1^ exhibited almost symmetrical isosceles triangles (Appendix A), implying their excellent electrochemical reversibility. The rate performances are compared in Appendix A, which were consistent with CV.

Figure 6a shows the CV curve of B/S-SC_S_-1 at different scan rates; it can be observed from this figure that the B/S-SC_S_-1 electrode exhibited a broad peak in the potential range of −1 to 0 V, which suggests that the existence of superficial hetero atoms in large numbers is conducive to the formation of significant pseudocapacitance. There are two possible causes for these phenomena. On the one hand, a greater number of active heteroatoms (especially B, S) not only increase pseudocapacitance but also expose more ions on the hydrophilic surface while expanding the area of the CV ring. On the other hand, the three-dimensional porous structure promotes the conduction of electrons and improves the charge transfer [45]. Figure 6b shows the GCD diagram of B/S-SC_S_-1 at varying current densities. At each level of current density, the constant current charge–discharge curve corresponded to the CV curve, despite a slight deformation and a quasi-symmetrical triangle, indicating the reliable reversibility and the existence of both EDLC and pseudocapacitance. In addition, the ratio capacitances of the B/S-SC_S_-1 electrode diminished with the elevation of the current density, as shown in Figure 6c, which results from the reinforced diffusional limitation and lesser ionic transport time under the increased current density [13]. B/S-SC_S_-1 showed a maximum ratio capacitance of 290.7 F g^−1^ at 0.5 A g^−1^ and exhibited an even higher capacitor retention of 70%, even at 10 A g^−1^, suggesting a remarkable performance. The electrochemical performance of B/S-SC_S_-1 is also extremely excellent compared with previous research works (Table 2). It can thus be inferred that the heteroatom-rich electrode can not only help enhance pseudocapacitance but also improve the contact probabilities between the electrode and the electrolyte, thus ensuring a large effective surface for charge storage. Ultimately, an excellent rate of performance can be achieved for the electrode.

B/S-SC_S_-1 performs well at a specific capacitance, which is attributable to the synergistic effect produced by various heteroatoms. Figure 7 illustrates the mechanism followed by the synergistic effect of B and S doping on the increased capacitance for B/S-SC_S_-1. According to a previous report [43], the oxygen-containing groups can improve wettability and pseudocapacitance (the possible redox reactions are shown in Figure 7). The nitrogen functional groups, where N acts as heteroatoms, could ensure steady pseudocapacitance, possibly ascribed to amine groups’ redox reactions [43], as shown in Figure 7. Additionally, the B heteroatom could be treated as an electrocatalyst to accelerate the oxidation-reduction reaction of oxygen and nitrogen functional groups, thus further improving pseudocapacitance [44,51]. Particularly, the substitutions of carbon with boron in the carbon materials lattice cause a downward shift to the Fermi level, thereby enhancing charge storage and transfer within the structure of doped carbon materials [52]. The contribution of the S atom to the electrochemical property could be attributed to the introduction of C-S, which enables the n-type dopant S to provide more reversible false sites and polarized surfaces [53]. In addition, sulfur-containing functional groups can be used as a support framework for larger pores at high temperatures to reduce the shrinkage of micropores during carbonization, which enables fast ion transmission (the possible groups are displayed in Figure 7) [2].

Figure 8 shows the electrochemical performance of the B/S-SC_S_-1//B/S-SC_S_-1 symmetric supercapacitor after the introduction of the 6 M KOH electrolyte(The real image of the symmetric device is presented in Appendix A). The CV curves of the prepared B/S-SC_S_-1//B/S-SC_S_-1 supercapacitor exhibit quasi-rectangular shapes with no significant deformation, even at a high scanning velocity of 200 mV s^−1^ (Figure 8a), which suggests a dominant electric double layer, fast charge/discharge process, and satisfactory electrochemistry rate property [54]. As shown in Figure 8b, the GCD curve exhibited a typical triangle, indicating high capacitance and electrochemical reversibility. From Figure 8c, it can be seen that the B/S-SC_S_-1//B/S-SC_S_-1 symmetrical supercapacitor assumed a high specific capacitance of 200 F g^−1^ at 0.5 A g^−1^, and when the current density increased to 10 A g^−1^, it reached a ratio capacitance of 116.4 F g^−1^, revealing moderate rate performance. The B/S-SC_S_-1/B/S-SC_S_-1 supercapacitor has a specific energy of 6.94 Wh kg^−1^ and specific power of 248.4 W kg^−1^ (Figure 8d). EIS analysis confirmed the enhanced kinetics of ionic transportation and charge transport in the constructed supercapacitor, as indicated by the Nyquist plot in Appendix A. The almost vertical lines in the low-frequency area demonstrate the typical electric double-layer capacitor (EDLC) behavior with fast ion diffusion and migration [55]. At high frequencies, the presence of a full, tiny semicircle substantiates the low charge transfer resistance during the Faraday reaction and EDLC creation [38].

To further explore the practical application of electrode materials in supercapacitors, a prototype of a two-electrode supercapacitor was produced using B/S-SC_S_-1 in neutral 1 M Na_2_SO_4_ solutions. Compared with acidic or alkaline electrolytes, Na_2_SO_4_ electrolyte can offer a large voltage window. Since hydrogen and oxygen evolution reactions occur as the voltage increases, this causes a rapid increase in the current at a high potential. The neutral electrolytic liquid phase has less H^+^ and OH^−^ for acidic and alkaline. In addition, the porous carbon structure provides rich transport and adsorption sites for H^+^ and OH^−^, thus inhibiting hydrogen and oxygen evolution reactions [56,57]. Figure 9a shows the CV measurement performed at 50 mV s^−1^ for the B/S-SC_S_-1/B/S-SC_S_-1 supercapacitor. When the voltage swing increases to 1.8 V, the CV curves basically remain rectangular and symmetrical, with no visible redox peaks, showing ideal capacitance behavior.

Figure 9b reveals the perfect symmetry and outstanding reversibility of the GCD curve. Based on the discharging time, the specific capacitances were estimated to be 142 F g^−1^ at the current density of 0.5 A g^−1^ (Figure 9c). The results show a downward trend with the increase in the current density. Based on the specific capacitance value and high-voltage operating range, Figure 9d shows the Ragone diagram of the B/S-SC_S_-1//B/S-SC_S_-1 symmetric supercapacitor. It exhibits maximum specific energy of 16.65 Wh kg^−1^ at a specific power of 450 W kg^−1^ in the B/S-SC_S_-1//B/S-SC_S_-1 symmetrical supercapacitor, which is considerably higher compared to the previously reported carbon-based symmetric supercapacitors (Table 3). The samples prepared using this method can achieve a more balanced and outstanding performance with respect to capacitance, specific energy, and specific power. The cycling stability of B/S-SC_S_-1 was investigated, as shown in Figure 10, and the capacitor retention of the B/S-SC_S_-1 electrode was around 97.09% posterior to 10,000 cycles at the current density of 10 A g^−1^. The results indicate that the prepared B/S-SC_S_-1 could produce an excellent longtime cyclic performance, it may be related to the variation of the doping content [58]. Appendix A presents the Nyquist plots of B/S-SC_S_-1//B/S-SC_S_-1. Apparently, the curve is presented as an almost-vertical line at low frequency, which implies the excellent performance of the electrochemical capacitor. Moreover, the smaller semicircle of B/S-SC_S_-1//B/S-SC_S_-1 suggests a high ionic diffusion–transference velocity [59].

To investigate the universality of this method, three biomass wastes, including walnut peel, wheat straw, and corn stalk, were employed to synthesize pore-rich carbon via an identical scheme. Appendix A show the electrochemical performances of the corresponding materials. Ratio capacitances of the pore-rich carbon acquired from the walnut peel, wheat straw, and corn stalks were 240, 223, and 200.2 F g^−1^ at 0.5 A g^−1^, respectively. Their voltage also reached 1.8 V in a symmetrical system of 1 M Na_2_SO_4_ and had a specific energy of 12 Wh kg^−1^, 12.5 Wh kg^−1^, and 11.3 Wh kg^−1^, respectively. Additionally, they exhibited brilliant electrochemical steadiness posterior to 10,000 cycles with 98.2%, 97.6%, and 100.67 % capacitor retention, respectively. These results emphasize the versatility of our scheme in the preparation of pore-rich carbon.

## 4. Conclusions

In summary, our team developed a simple and versatile method to prepare B/S co-doped porous carbon. Due to the synergistic effect between the activation treatment of ZnCl_2_ and the doping of B and S, the as-prepared B/S-SC_S_-1 as-electrode material displayed a great capacitance of 290.7 F g^−1^ at 0.5 A g^−1^ as well as a brilliant rate performance. In addition, the supercapacitor established with B/S-SC_S_-1 produced maximal specific energy of 16.65 Wh g^−1^ in the 1 M Na_2_SO_4_ electrolyte, and it showed high stability with 97.09% of the initial capacitance retention posterior to 10,000 cycles. Remarkably, boric acid and thiourea proved to be effective sources of heteroatom doping. Furthermore, we used the same scheme to achieve the effective preparation of electrode materials for the other three biomass wastes. Considering the availability and versatility of the method, we expected that the B and S-codoped porous carbons materials should have broad application prospects in energy-storage devices. In the long-term sense, the full utilization of these biomass wastes can not only realize the idea of turning waste into treasure but also solve the energy problem.

## Figures and Tables

**Figure 1 nanomaterials-12-01182-f001:**
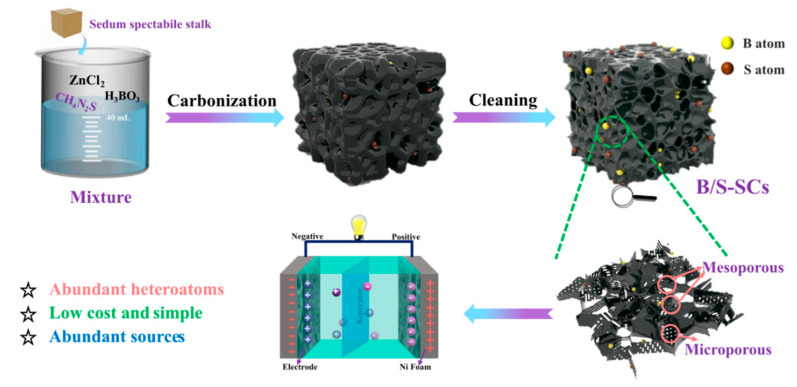
The synthesis route of B/S-SCs electrode.

**Figure 2 nanomaterials-12-01182-f002:**
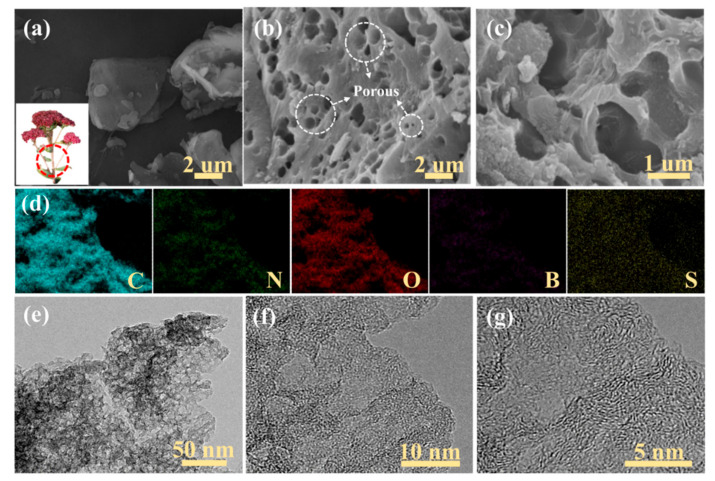
(**a**) SEM images and a photograph (inset) of SC; (**b**,**c**) SEM images of SC; (**d**) element mapping of C, N, O, B, and S in B/S-SC_S_-1; (**e**–**g**) TEM images of B/S-SC_S_-1 at different magnifications.

**Figure 3 nanomaterials-12-01182-f003:**
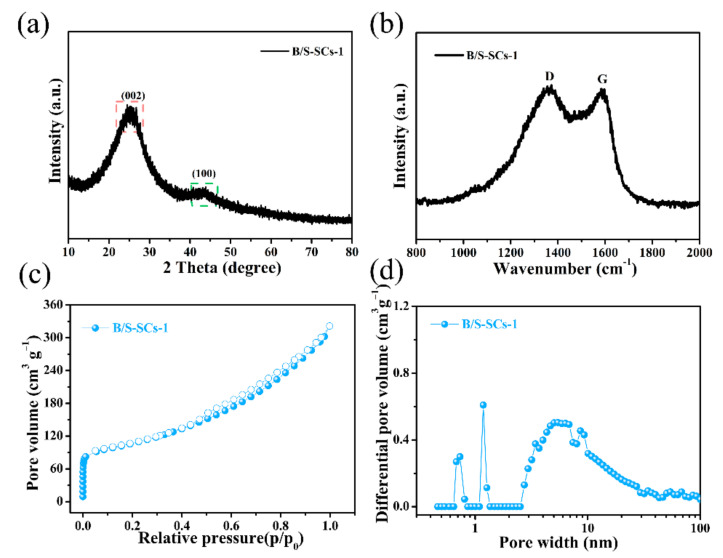
(**a**) XRD pattern; (**b**) Raman spectra; (**c**) nitrogen adsorption/desorption isotherm; (**d**) pore size distribution of the obtained B/S-SC_S_-1 electrode.

**Figure 4 nanomaterials-12-01182-f004:**
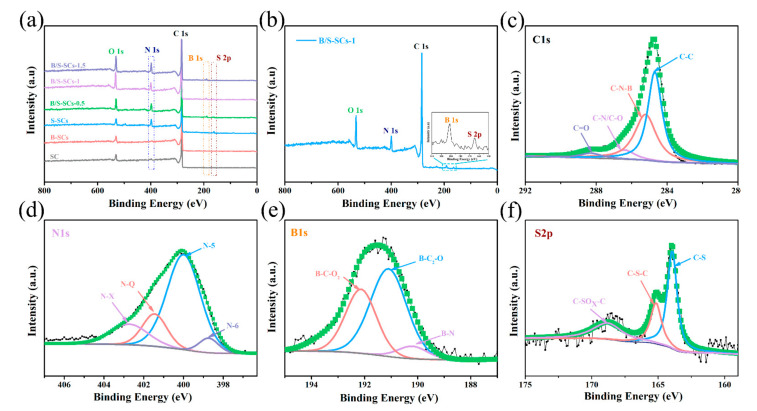
(**a**) XPS spectra of all samples; (**b**) XPS spectra of the B/S-SC_S_-1; (**c**) high-quality C 1s XPS images of B/S-SC_S_-1; (**d**) high-quality N 1s XPS images of B/S-SC_S_-1; (**e**) high-resolution B 1s XPS images of B/S-SC_S_-1; (**f**) high-quality S 2p XPS images of B/S-SC_S_-1.

**Figure 5 nanomaterials-12-01182-f005:**
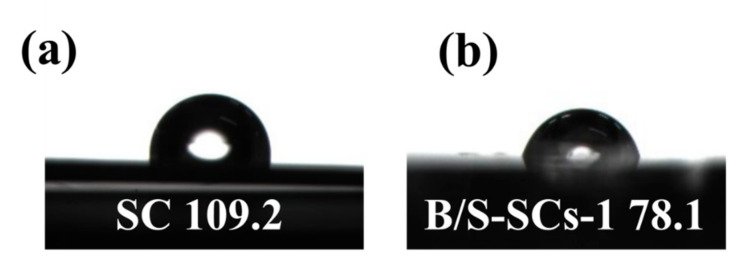
The contact angles formed by water drops on the surface of (**a**) SC and (**b**) B/S-SC_S_-1.

**Figure 6 nanomaterials-12-01182-f006:**
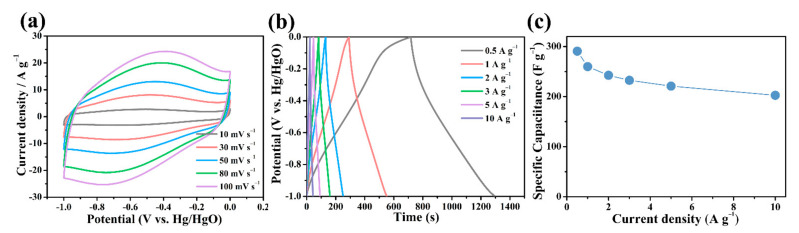
(**a**) CV and (**b**) GCD curves of B/S-SC_S_-1 at the different scan rates and current densities; (**c**) gravimetric capacitance as a function of specific current.

**Figure 7 nanomaterials-12-01182-f007:**
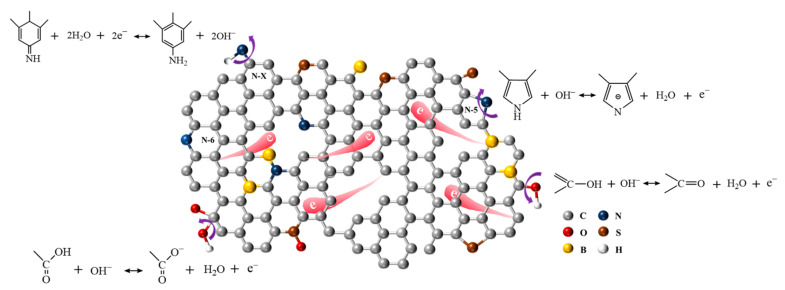
Effects of the N, O, B, and S heteroatoms on the increased capacitance of B/S-SC_S_-1.

**Figure 8 nanomaterials-12-01182-f008:**
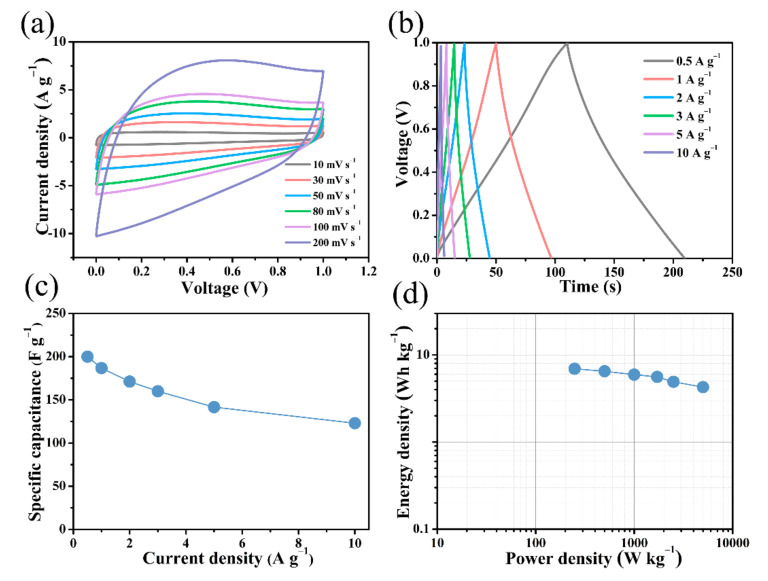
Electrochemical performance of B/S-SC_S_-1//B/S-SC_S_-1 symmetric supercapacitor using 6 M KOH electrolyte. (**a**) CV curves at various scanning rates; (**b**) GCD curves under various ampere densities between 0.5 and 10 A g^−1^; (**c**) gravimetric capacitance as a function of specific current; (**d**) Ragone plots.

**Figure 9 nanomaterials-12-01182-f009:**
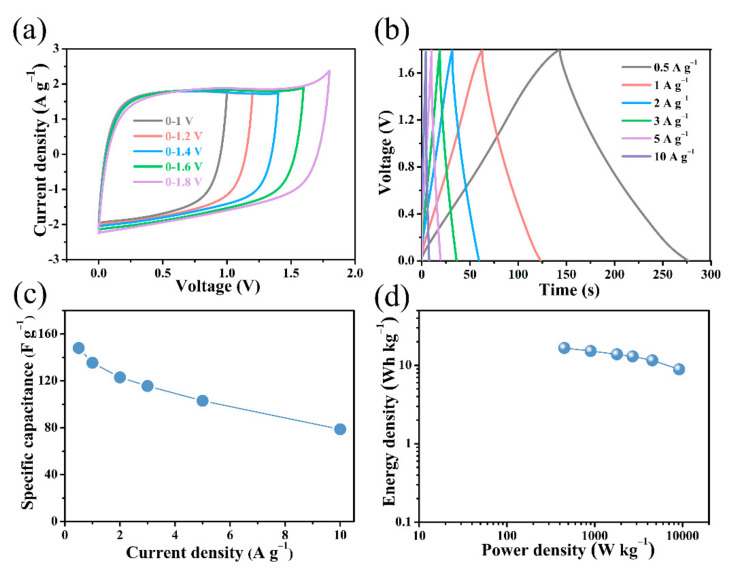
Electrochemical performance of B/S-SC_S_-1//B/S-SC_S_-1 symmetric supercapacitor via 1 M Na_2_SO_4_ electrolyte. (**a**) CV curves acquired voltage transformation between 1 and 1.8 V under a scanning velocity of 50 mV s^−1^; (**b**) GCD curves under various current densities between 0.5 and 10 A g^−1^; (**c**) gravimetric capacitance as a function of specific current; (**d**) Ragone plot.

**Figure 10 nanomaterials-12-01182-f010:**
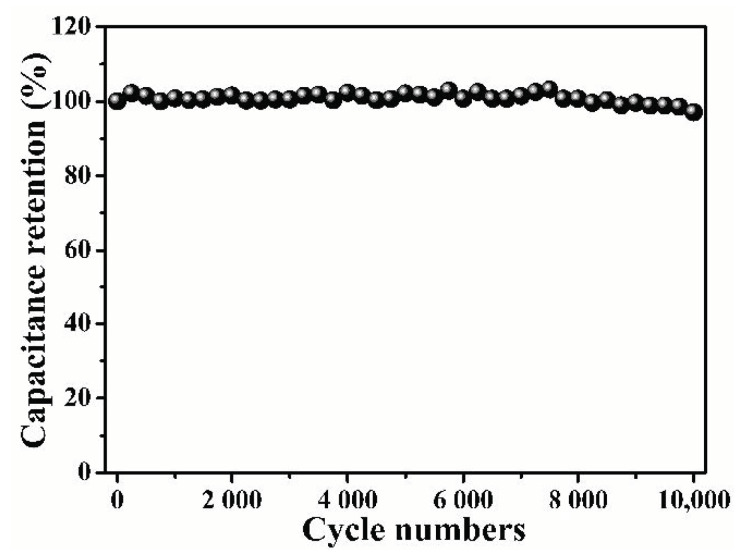
Cycling stability of the B/S-SC_S_-1//B/S-SC_S_-1 symmetric supercapacitor using 1 M Na_2_SO_4_ electrolyte at 10 A g^−1^.

**Table 1 nanomaterials-12-01182-t001:** The element contents in the as-prepared samples determined by XPS.

Materials	Element Content (at.%)				
	C	N	O	B	S
B/S-SCs-1	72.75	10.5	10.99	5.56	0.81

**Table 2 nanomaterials-12-01182-t002:** Comparison of specific capacitance of B/S-SCs-1 with the reported biomass-derived carbon materials in references.

Carbon Source	Electrolyte	Specific Capacitance (F g^−1^)	Current Density (A g^−1^)	Refs.
Platanus	6 M KOH	286 F g^−1^	0.5 A g^−1^	[46]
Composting leachate	6 M KOH	228 F g^−1^	0.5 A g^−1^	[47]
Pork bone	6 M KOH	302 F g^−1^	0.5 A g^−1^	[48]
Mulberry leaves	6 M KOH	214.5 F g^−1^	0.5 A g^−1^	[49]
Licorice root	6 M KOH	221 F g^−1^	0.5 A g^−1^	[50]
Sedum spectabile stalk	6 M KOH	290.7 F g^−1^	0.5 A g^−1^	This work

**Table 3 nanomaterials-12-01182-t003:** Comparison of the specific energy value of the supercapacitor with biomass carbon in recent references.

Carbon Source	Electrolyte	Specific Energy	Refs.
Cashew nut husk	1 M Na_2_SO_4_	11.2 Wh kg^−1^	[19]
Pine nut shells	1 M Na_2_SO_4_	11.9 Wh kg^−1^	[60]
Perilla frutescens leaves	1 M Na_2_SO_4_	13.9 Wh kg^−1^	[61]
Carrot	1 M Na_2_SO_4_	13.9 Wh kg^−1^	[62]
Sedum spectabile stalk	1 M Na_2_SO_4_	16.65 Wh kg^−1^	This work
Walnut peel	1 M Na_2_SO_4_	12 Wh kg^−1^	This work
Wheat straw	1 M Na_2_SO_4_	12.5 Wh kg^−1^	This work
Corn stalks	1 M Na_2_SO_4_	11.3 Wh kg^−1^	This work

## Data Availability

The data presented in this study are available on request from the corresponding author.

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
