# Peer review of "Preparation of Boron/Sulfur-Codoped Porous Carbon Derived from Biological Wastes and Its Application in a Supercapacitor"

_nanomaterials, 2022, doi:10.3390/nano12071182_

Round 1
Reviewer 1 Report
In the work, the authors report a strategy to fabricate boron/sulfur co-doped porous carbon from biomass sources for the supercapacitor application. The obtained carbon as electrode material shows specific capacitances of 290.7 F g-1 at 1 A g-1 current density. Moreover, the energy density of the prepared samples as-assembled symmetric supercapacitor is 16.65 Wh kg-1in 1 M Na2SO4. After addressing the following comments this article can be accepted in this journal.
- This short form (B/S-SCS-1) needs to explain before its use in the introduction section.
- Authors write specific capacitances of 290.7 F g-1 at 1 A g-1 in abstract but in line 79 this is 290.7 F g-1 at 0.5 A g-1, it needs to be corrected!
- For the three-electrode system authors use 6M KOH as an electrolyte but for the two-electrode system (symmetric supercapacitor) 1 M Na2SO4 was used. Why 6M KOH is not used in the device fabrication? What is the necessity and significance of the different electrolytes for two different systems? I suggest using either 6M KOH or 1M Na2SO4, but the same electrolyte for both systems.
- What is the reason behind the increased potential window of Symmetric supercapacitor B/S-SCS-1//B/S-SCS-1 while using 1 M Na2SO4. Explain the mechanism in results and discussion.
- From Figure 2 (d), the corresponding EDX should be added. Wt% and Atomic % of all the elements should be shown clearly.
- The real image of the raw materials should be added to Figure 2.
- In figure 1, the thrombolysis step also needs to show clearly by redesigning the schematic.
- In the experimental section authors should indicate the concentration of HCl.
- In figure 5, the authors forget to label (a), (b), (c), etc. It needs to be corrected. Furthermore, in the second figure, B1s and S2p should be indicated.
- Ragone plot should be added to Figure 7 with comparison to the similar type of reported literature. (Figure S6 (c) is not sufficient).
- In supporting information authors include the corn stalk derived porous carbon, wheat straw derived porous carbon, and walnut peel derived porous carbon. What is the significance of all these materials' data presentations in supporting info? Furthermore, the authors do not include the preparation method of such materials and are not explained in the result and discussion section.
- The EIS data presented in Figure S4 should be fitted to calculate the Rct and Rs values. Furthermore, the EIS of the Symmetric supercapacitor device should add.
- The authors should add two different tables for the comparison of similar types of materials from reported literature in a 3-electrode system and symmetric supercapacitor.
- The presentation of the data should be rearranged. Some figures can be combined, (for example Figures 3 and 4). some of the recent work of supercapacitors can be cited and taken as a data presentation reference as indicated below:10.1016/j.carbon.2021.04.028 and 10.1016/j.jelechem.2019.113670.
- What type of symmetric device was fabricated? The real image should be included in the Supporting info.
- What is the novelty of this work? It should be elaborated in the last paragraph of the introductions.
- In figure 6, authors should compare the EIS of all the comparable samples.
Reviewer 2 Report
Reviewer comments
The paper reports on the synthesis and characterization of boron/sulfur-doped carbons, derived from biological wastes, for potential application in double-layer electrochemical capacitors. When it comes to characterization methods and results, I found this paper a nice piece of work. However, I have some concerns about the novelty of the research. Nowadays, there is a growing number of papers describing carbon materials derived from bio-wastes. Some of these materials are doped with single heteroatoms, but there are also many reports co-doped materials. The authors write that most of doping works on carbon concern single-doping processes. I cannot agree. As far as I know, there are plenty of papers about co-doped carbons, eg. boron-nitrogen, nitrogen-phosphorous or nitrogen-oxygen. Indeed, co-doping with sulphur and other atoms is rather rare (although some papers describing nitrogen-sulphur but also boron-sulphur carbon materials exist). For instance, please take a look at these references:
https://doi.org/10.1016/j.apcatb.2021.120144
https://doi.org/10.1142/S1793292017501235
doi: 10.20964/2018.04.37
Therefore, I think the reason for conducting such studies should be emphasized, especially in the Introduction section which should be particularly rewrtitten. The authors should convinced the Reader to the purpose and necessity of the research. The advantages of the approach, and/or this particular material’s properties (in comparison to similar, already reported materials) should be underlined as well. Moreover, I have some further important comments and questions that are listed below:
- Introduction, line 61: “hydrophobicity” – rather “hydrophobic”
- Line 78: “super capacitor”, it should be “supercapacitor”. The same refers to “pseudocapacitance”. Please check carefully the whole text (including Suppl. Materials) to find all clumsy/improper words, sentences with bad construction, missed, or unnecessary, spaces, etc.
- I would be careful about the voltage window of the cell in neutral (Na2SO4) electrolyte which was determined to be 1.8 V. There is a number of papers by prof. F. Beguin at al. showing various ageing phenomena of supercapacitors operating in neutral salt electrolytes occurring at voltage >1.5 V. They employed a “floating” technique, instead of a typical constant current charge-discharge over number of cycles, to discover a real long-term operation of the systems. I encourage to use such an approach in future studies for all cells operating at elevated voltages. Especially that in the paper a high specific current was applied in the cycling procedure (10 A/g, Fig. S5). So, the cycleability test was not that long in practice. Hence, it is difficult to comment on practical stability (supercapacitors are destined to operate over 106 of cycles. Aforementioned floating procedure simulates such conditions during 100-200 h of floating period.). I also recommend to move the cycleability test to the main body of the manuscript. Despite all I’ve written, it’s important result.
- Geometric area, mass and thickness of the carbon electrodes used for capacitive properties and evaluation (in both (2-, 3-) electrode configuration) should be given.
- I think more experimental/calculation details is needed for N2-sorption analysis. For instance, what was the model used for the PSD calculations? How the BET surface area was determined exactly? What was the outgassing time/temperature? What was the sample mass? What was the pore total/micro/meso volume/area and the average pore size? Also, at the isotherm (Fig. 4a), I cannot see any points at low p/p0. Does it mean the material has no micropores or they have not been measured? I think the latter is correct which means the measurement is incomplete.
- The authors refer to the difference in the charge-discharge characteristics of different single-doped, co-doped or un-doped carbon materials but the physicochemical characteristics are nicely done only for one of them (for the one which shows the best capacitive properties). However, it would be interesting to know what are the morphological/structural/textural differences between all materials employed into supercapacitor in a case they are mentioned and examined. Otherwise, I don’t see any sense to show the electrochemical; data for various carbons, especially that I don’t see many comments on the observed differences. Maybe the idea is to put Fig. 6 to the SupplementarySection.
- Figure captions, eg. Fig. 6c “Ratio capacitances of all samples under diverse ampere densities”. It looks a little bit akward. Maybe: “Gravimetric capacitance as a function of specific current”?. By the way, current normalized per mass should not be called current density but rather specific current or mass normalized current. Current density means normalization per electrode area or volume.
- The authors suggest that highly hydrophilic surface of B/S SC1 material is responsible for its superior electrical capacitance, in addition to pseudocapacitance effect of heteratoms. Is it possible to conduct, eg. contact angle measurements to compare the wettability of investigated samples?
- Caption (Fig. 7)– the figures are numbered from a to d which is not in agreement with caption to Fig. 7.
- Caption (Fig. 10): “symmetry super capacitor”, it should be “symmetric supercapacitor”. Please, revise the whole text thoroughly.
- The EIS spectra in Fig. S4 are of low quality. It is hard to see the numbers in the inset even upon enlarging the plot (low resolution)
- In Fig, 7d there is a comparison of reported data to literature data. Do all these already reported data refer to the same (i.e. 3-electrode) electron configuration and the same electrolyte? In other case, such comparison would be meaningless. That should be clarified.
- Are the authors able to give some details about the electrochemical setup (2- and 3-electrode configuration)? I’m mainly interested in the 3-electrode configuration and the distance between working and reference electrodes. It’s because the authors report on the series resistance equal to 0.7 Ohm in KOH electrolyte. This value is strictly related to the geometry of the electrode (mass loading, area, thickness) but also to the distance between the aforementioned electrodes. In case of a significant distance and the absence of Luggin capilary, the ohmic compensation should be applied which is a challenge using the CHI Instrument. Therefore, it is always better to comment on the resistance contributions using two-electrode cell data.
- Page 9, line 280: “watt density” - it should be a specific power, “energy density” – it should be specific energy.
- The symmetric capacitor in KOH electrolyte retains 60% of the specific capacitance when specific current is increased from 0.5 to 10 A/g. For me it is not a superior behavior, rather moderate which is not surprising due to the mixed double layer/faradaic charge storage mechanism and the kinetic limitations of the latter at higher rates. The comparison of rate behavior between reported and already published data is also presented in Fig. 7d. Please be careful with that. Such comparison makes sense only when the compared results correspond to the identical cells, i.e. only the electrode material is different but other important factors are the same (i.e. electrolyte, electrode geometry, separator type and thickness, and the force applied to the electrodes during assembly in the electrochemical cell, eg. using the compression spring of particular type).
- 10d: The comparison of reported data with the literature results on Ragone plot: the comparison makes sense only when all presented data correspond to the neutral electrolyte-based supercapacitors. Can the authors confirm that?
On the whole, I think the manuscript needs deep major revision by taking into accounts my suggestions. Also, the language and style of presentation must be improved; especially the electrochemical part should be revised very carefully.
Round 2
Reviewer 1 Report
All the comments are address except EIS related questions.
Author Response
Thank you very much for your hard work on our manuscript.
Reviewer 2 Report
I would like to thank for all answers to my comments, and for the preparation of a comprehensive revision of the manuscript. I have just few more minor suggestions:
- Fig. 10: The Fig. presents the capacitance retention vs the number of cycle. I cannot see any coulombic efficiency data mentioned in the fig. caption (which is as a rule defined as Qin/Qout). Please remove "coulombic efficiency" from the figure description or include appropriate data.
- Table 3:"Comparison of the specific energy value of the sample with biomass carbon in recent references." Please replace "Sample" with "supercapacitor" for clarity
- The electrochemical part still contains some gramatical/typographical and nomenclature errors. For instance: "electrochemistry capacitor=electrochemical capacitor, produuved=produced, ampero density=current density, electric pole=electrode, in ratio=with respect to. Spaces are sometimes missing. Please check the whole part carefuly.
- To summarize, I recommend publication of the article after minor revision.
